# Intra- and interspecific diversity in a tropical plant clade alter herbivory and ecosystem resilience

Ari Grele[1], Tara J Massad[2], Kathryn A Uckele[1], Lee A Dyer[1,3], Yasmine Antonini[4], Laura Braga[4], Matthew L Forister[1,3], Lidia Sulca[5], Massuo Kato[6], Humberto G Lopez[1], André R Nascimento[7], Thomas Parchman[1,8], Wilmer R Simbaña[9], Angela M Smilanich[1], John O Stireman[10], Eric J Tepe[11], Thomas Walla[12], Lora A Richards[1,3]*

[1]Program in Ecology, Evolution, and Conservation Biology, Department of Biology, University of Nevada, Reno, United States; [2]Department of Scientific Services, Gorongosa National Park, Sofala, Mozambique; [3]Hitchcock Center for Chemical Ecology, University of Nevada, Reno, United States; [4]Lab. de Biodiversidade, Departamento de Biodiversidade, Evolução e Meio Ambiente, Instituto de Ciências Exatas e Biológicas, Universidade Federal de Ouro Preto, Ouro Preto, Brazil; [5]Departamento de Entomología, Museo de Historia Natural, Universidad Nacional Mayor de San Marcos, Lima, Peru; [6]Department of Fundamental Chemistry, Institute of Chemistry, University of São Paulo, São Paulo, Brazil; [7]Department of Ecology, Universidade Federal de Goiás, Goiânia, Brazil; [8]Department of Biology, University of Nevada, Reno, United States; [9]Yanayacu Biological Station, Cosanga, Ecuador; [10]Department of Biological Sciences, Wright State University, Dayton, United States; [11]Department of Biological Sciences, University of Cincinnati, Cincinnati, United States; [12]Department of Biology, Mesa State College, Grand Junction, United States

*For correspondence: lorar@unr.edu

**Abstract** Declines in biodiversity generated by anthropogenic stressors at both species and population levels can alter emergent processes instrumental to ecosystem function and resilience. As such, understanding the role of biodiversity in ecosystem function and its response to climate perturbation is increasingly important, especially in tropical systems where responses to changes in biodiversity are less predictable and more challenging to assess experimentally. Using large-scale transplant experiments conducted at five neotropical sites, we documented the impacts of changes in intraspecific and interspecific plant richness in the genus *Piper* on insect herbivory, insect richness, and ecosystem resilience to perturbations in water availability. We found that reductions of both intraspecific and interspecific *Piper* diversity had measurable and site-specific effects on herbivory, herbivorous insect richness, and plant mortality. The responses of these ecosystem-relevant processes to reduced intraspecific *Piper* richness were often similar in magnitude to the effects of reduced interspecific richness. Increased water availability reduced herbivory by 4.2% overall, and the response of herbivorous insect richness and herbivory to water availability were altered by both intra- and interspecific richness in a site-dependent manner. Our results underscore the role of intraspecific and interspecific richness as foundations of ecosystem function and the importance of community and location-specific contingencies in controlling function in complex tropical systems.

## eLife assessment

This **important**, large experimental study examines the effects of plant species richness, plant geno-typic richness, and soil water availability on herbivory patterns for Piper species in several tropical sites. The authors find **solid** evidence that water availability, as well as intra- and interspecific plant diversity, influence herbivory and herbivore diversity, but that the effects differ geographically.

## Introduction

As climate change and anthropogenic activity alter ecosystems at unprecedented rates, it has become critical to understand the consequences of biodiversity loss on ecosystem processes and the mainte-nance of ecosystem processes through species interactions. A complex mix of anthropogenic forces are eroding multiple dimensions of global biological diversity, including plant intraspecific, interspe-cific, and functional diversity (**Morris, 2010**; **Oliver and Morecroft, 2014**). Plant diversity affects herbivore abundance and diversity, thereby influencing biomass allocation and energy fluxes between trophic levels (**Ebeling et al., 2018**). Because losses of plant diversity can destabilize the flow of resources to higher trophic levels (**Ebeling et al., 2018**; **Naeem and Li, 1997**), understanding the connection between biodiversity and trophic interactions is necessary to predict the consequences of the loss of primary producer biodiversity on ecosystem traits such as resilience to environmental perturbation (**Elmqvist et al., 2003**; **Oliver et al., 2015**). While climate change is associated with increased absolute precipitation in some regions and decreased precipitation in others, IPCC models predict mid-century increases in the frequency of extreme precipitation events in Central and South America (**Easterling et al., 2000**; **O'Gorman and Schneider, 2009**; **Field et al., 2014b**), a phenom-enon already observable in many ecosystems (**Fischer and Knutti, 2016**). Measuring how precipi-tation change will affect the relationship between biodiversity and ecosystem function is therefore increasingly important, particularly in diverse tropical systems.

Very few multi-site, manipulative diversity experiments have been reported from tropical areas compared to temperate environments (**Clarke et al., 2017**), limiting our knowledge of the role of biodiversity in ecosystem function in the most species-rich regions of the planet (**Gentry, 1992**). In the context of several established hypotheses (**Table 1**), we investigate how multiple dimensions of plant diversity affect ecosystem processes at five neotropical sites and explore how diversity modu-lates how ecosystems respond to changes in water availability at three of those sites. We focus on ecosystem responses that represent changes in energy fluxes between trophic levels as measured by herbivory, herbivore diversity, and plant mortality. By altering plant uptake of nutrients and plant defense production, abnormal levels of precipitation can alter herbivore pressure, affecting the move-ment of resources into higher trophic levels (**White, 1974**). As such, extreme dry and wet periods of climate are expected to strongly perturb plant-insect interactions and thereby alter ecosystem function

**Table 1.** Path models and explanatory hypotheses.

Causal path labels refer to *Figure 3—figure supplement 2*.

| Hypothesis | Causal paths | Sites tested |
|---|---|---|
| **Bottom-up diversity:** Host plant diversity impacts insect richness through multiple mechanisms, such as reducing host density, masking or amplifying host signals, or altering the proportion of specialist herbivores (**Agrawal et al., 2006**; **Barbosa et al., 2009**; **Root, 1973**) | All models; paths A, C | All sites |
| **Neighborhood effects:** Plant diversity directly affects herbivory through mechanisms which do not alter herbivore diversity (**Agrawal et al., 2006**; **Barbosa et al., 2009**) | All models; paths B, D | All sites |
| **Water affects herbivore diversity:** Changes in water availability induce changes to plant nutrition and defenses which can benefit or harm different herbivore taxa, leading to changes in herbivore diversity (**Gely et al., 2020**; **Lenhart et al., 2015**) | Models I, II; path G | Costa Rica, Ecuador, Peru |
| **Water affects herbivory:** Water addition directly affects plant physiology, altering both the nutritive quality of plant tissue and the ability of plants to combat herbivores and leading to changes in herbivory (**White, 1974**) | Models I, III; path F | Costa Rica, Ecuador, Peru |

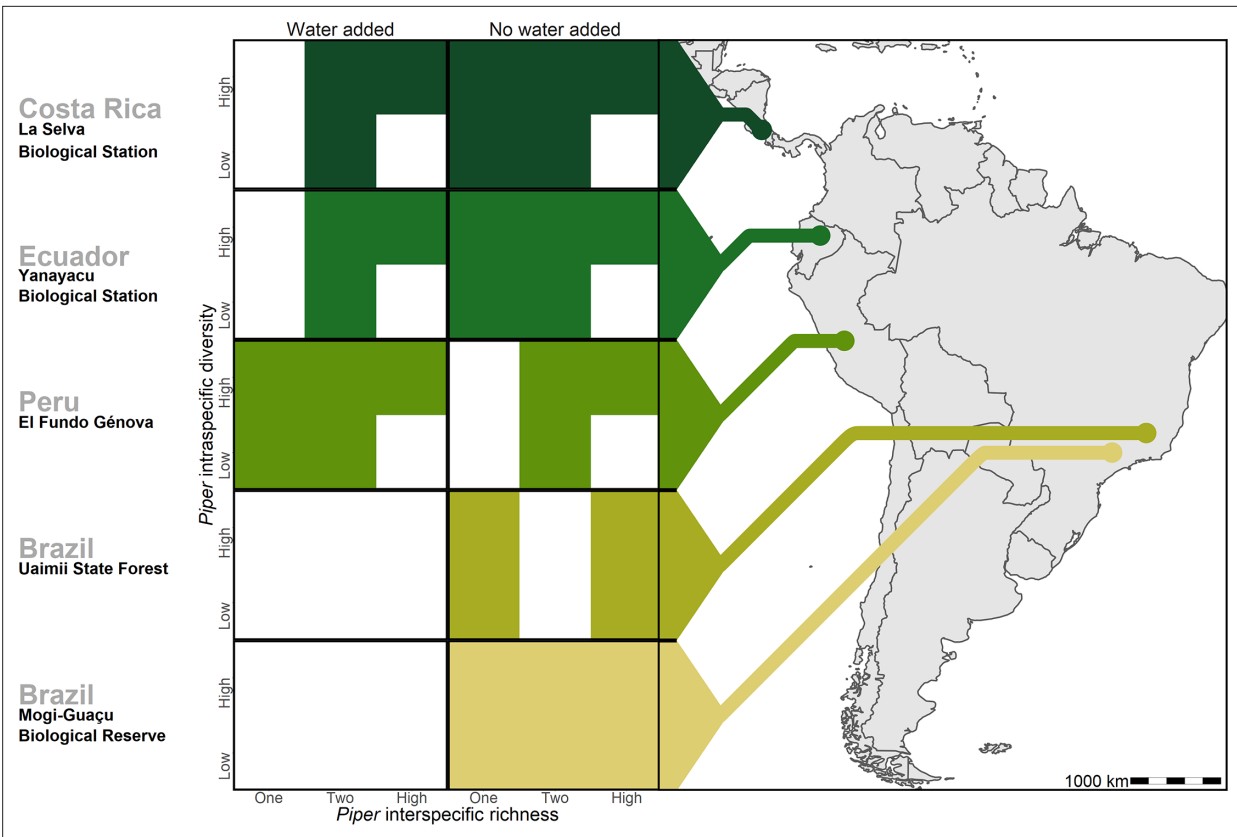

**Figure 1.** Treatments of intraspecific richness, interspecific richness, and water addition used in each of the five study sites. White tiles represent treatment combinations which were not tested in a given site.

The online version of this article includes the following figure supplement(s) for figure 1:

**Figure supplement 1.** Treatments and number of plots used across sites.

**Figure supplement 2.** Overall herbivory, plant mortality, and insect richness at five study sites.

**Figure supplement 3.** Precipitation levels at study sites where the water addition treatment was applied.

(**White, 1974**; **Côté and Darling, 2010**; **Koricheva et al., 1998**). The insurance hypothesis suggests that greater biodiversity can act to stabilize ecosystems and improve their resilience to environmental change (**Naeem and Li, 1997**; **Yachi and Loreau, 1999**). While greater interspecific plant richness is expected to lead to increased diversity in higher trophic levels due to the accumulation of specialist herbivores, field studies have demonstrated both positive and neutral effects of interspecific plant richness on ecosystem resilience (**Klaus et al., 2016**; **Lanta et al., 2012**). Despite traditional views that interspecific richness has a greater impact on ecosystem processes than intraspecific diversity (**Hughes et al., 2008**), recent research suggests that plant intra- and interspecific richness can have similar effects on ecosystem productivity and consumer abundance (**Raffard et al., 2019**; **Koricheva and Hayes, 2018**). As changes to intraspecific richness can alter the diversity of resources available to herbivores (**Crutsinger et al., 2006**), it is necessary to consider both inter- and intraspecific diversity when investigating the effects of biodiversity loss on ecosystem processes.

We conducted common garden experiments at five sites across Central and South America to test the insurance hypothesis by quantifying (1) the relative strength of intra- and interspecific plant richness in driving ecosystem function and (2) the effects of increased water availability on ecosystem function. Using 33 species in the genus *Piper* (Piperaceae) as a model system (**Dyer and Palmer, 2004**), we manipulated intra- and interspecific plant richness in Costa Rica, Ecuador, Peru, and two sites in Brazil. We additionally manipulated water availability in Costa Rica, Ecuador, and Peru (**Figure 1**). We predicted that reduced *Piper* diversity would lead to reduced diversity of higher trophic levels, that water addition would lead to altered herbivore pressure, and that lower *Piper* diversity would be associated with more extreme changes in herbivory and plant mortality in response to water

addition. Finally, we predicted that changes in intraspecific and interspecific plant richness would affect ecosystem processes, including herbivory, herbivore diversity, and plant mortality, with similar magnitudes.

## Results

Considerable variation in herbivory and plant mortality was observed among study sites. Percent herbivory was lowest at Mogi where only 9% of leaf tissue was consumed by herbivores compared to 22% of leaf tissue consumed in Uaimii. *Piper* mortality was highest in Peru (89%), likely due to El Niño related drought, and was lowest in Ecuador (27%; *Figure 1—figure supplement 2A*). We identified herbivore damage from 13 insect taxa at the five study sites, as well as damage by leaf miners of unknown orders. Damage from a total of 10 taxa were observed in Costa Rica, Ecuador, and Peru, 9 taxa were observed in Mogi, and 8 taxa were observed in Uaimii (*Figure 1—figure supplement 2B*). The majority of taxa in Peru, Mogi, and Uaimii were generalist herbivores, while 60% of taxa in Costa Rica and 58% of taxa in Ecuador were *Piper* specialists. The proportion of leaf tissue consumed by specialists was only greater than generalist damage in Costa Rica.

Our experiments revealed pronounced heterogeneity in ecosystem responses to water availability and *Piper* diversity between sites (*Figure 2*). Posterior predictive checks (PPCs) for all hierarchical Bayesian models (HBMs), and for models I and III were within 0.03 of 0.5, indicating models fit well. Model III (*Figure 3—figure supplement 2*) was selected as the most parsimonious causal model for Bayesian structural equation models (BSEMs) in Costa Rica (PPC = 0.499), Ecuador (PPC = 0.499), and Peru (PPC = 0.498). Fit was high for models in Mogi (PPC = 0.5) and Uaimii (PPC = 0.497). Across all sites where the water addition treatment was applied, percent herbivory was 4.2 ± 3.6% (mean ± 95% CI) lower in plots that received additional water (probability of direction [PD] = 98.7%; *Figure 2A and B*). Greater *Piper* interspecific richness was associated with a 15 ± 18.6% increase in the richness of insect herbivores (PD = 95.0%; *Figure 2E*) and an indirect increase in herbivory was mediated by insect richness (*Figure 3*). Insect richness was associated with an 8.8 ± 2.8% increase in herbivory per insect taxon present (PD = 100%) and a 6.7 ± 6.9% increase in the percentage of leaves with any damage (p=97%) (*Figure 2A and B*). Intra- and interspecific richness affected herbivory, although effects varied in strength and direction across sites (*Figure 3*, *Supplementary file 1*). Intraspecific richness had similar or greater effects on plant mortality and insect richness than interspecific richness. However, intra- and interspecific richness often had contrasting directions of effect on insect richness and measures of herbivore pressure (*Figures 2 and 3*). For example, in Costa Rica insect richness was 6.0 ± 5.9% (PD = 93.7%) lower in plots with high interspecific plant richness, while high intraspecific richness increased insect richness by 7.2 ± 6.9% (PD = 98.1%). In contrast, in Ecuador high interspecific richness was associated with a 43.6 ± 6.4% (PD = 100%) increase in insect richness, and high intraspecific richness decreased insect richness by 13.6 ± 5.2% (PD = 100%; *Figure 2—figure supplement 2B and C*).

The effects of water addition were altered by *Piper* intra- and interspecific richness at all sites (*Figure 4*). Water availability reduced herbivory in Costa Rica (4.7 ± 2.5%, PD = 100%) and Peru (5.1 ± 3.7%, PD = 100%), but this effect was only present in Ecuador when intraspecific richness was high (*Figure 3—figure supplement 1*). Across sites, water addition had negligible or negative effects on insect richness at low interspecific richness, but this pattern was reduced or reversed when interspecific richness was high. Insect richness increased by 20.8 ± 18.9% more in interspecifically diverse water plots compared to unwatered plots in Costa Rica (PD = 99%). Insect richness increased by 15.2 ± 16.8% more in rich, watered plots in Ecuador (PD = 96%), and 29.3 ± 31.5% more in Peru (PD = 98%). Additionally, water addition had a negligible effect on insect richness in Ecuador when intraspecific richness was low, but increased insect richness by 12.1% when intraspecific richness was high (PD = 99%; *Figure 4B*).

Water addition had a negative effect on *Piper* survival in Costa Rica when intraspecific richness was low, but improved survival by 12.1 ± 8.5% in high intraspecific richness plots (PD = 99%; *Figure 4C*). Interactions with water in Peru may have been influenced by an El Niño related drought which resulted in high *Piper* mortality, while the typically wetter sites in Costa Rica and Ecuador experienced greater precipitation than average (*Figure 1—figure supplement 3*).

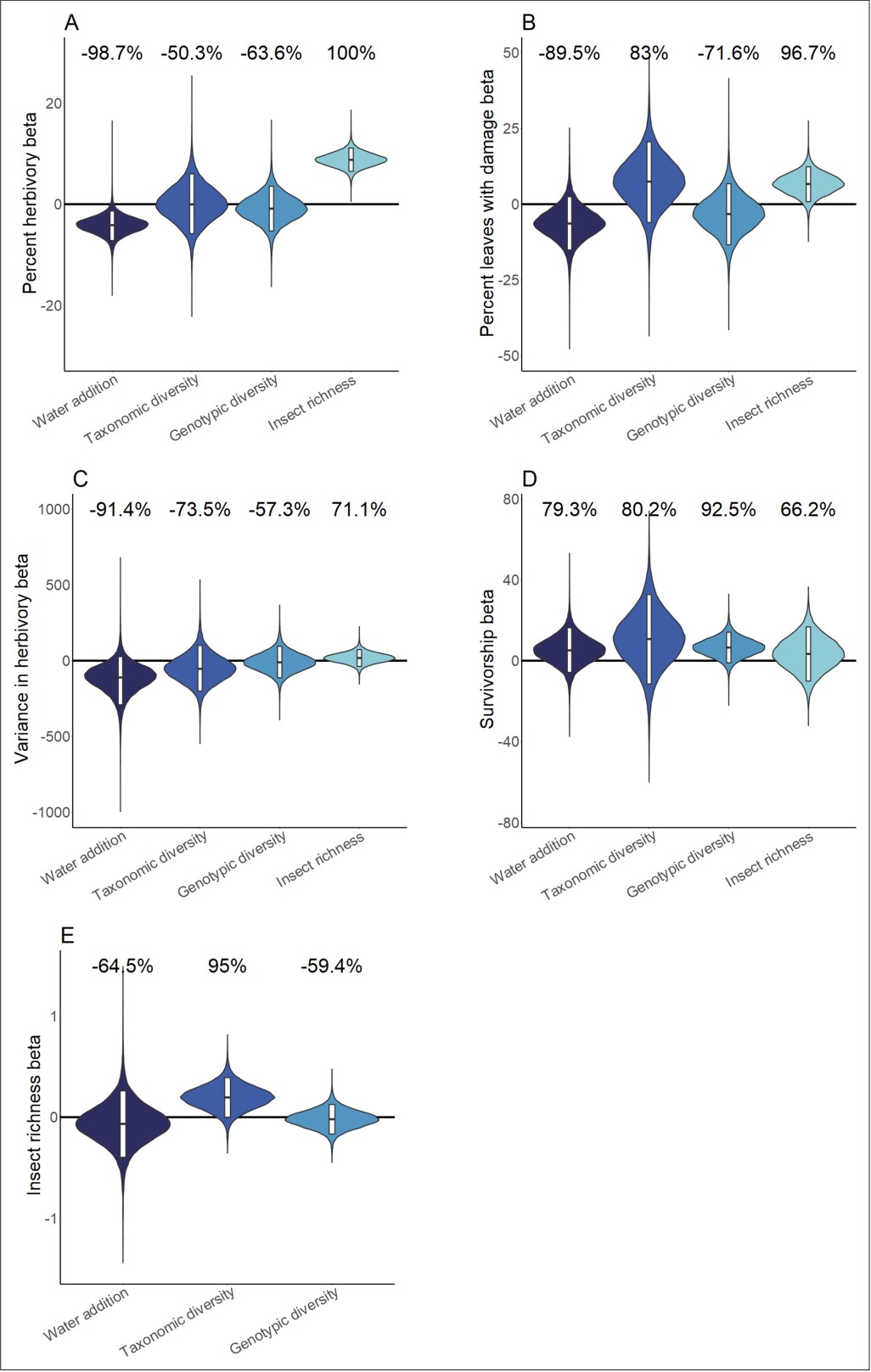

**Figure 2.** Hierarchical Bayesian model parameter estimates for the effects of water availability, as well as intraspecific and interspecific *Piper* richness on measures of herbivory (**A-C**), *Piper* survivorship (**D**), and herbivorous insect richness (**E**). Violins represent the cross-site posterior parameter distribution for each relationship in site-level hierarchical Bayesian models. Black lines represent the median posterior estimate and white bars represent

*Figure 2 continued on next page*

*Figure 2 continued*

95% credible intervals. Percentages above violins indicate the probability of an effect being positive or negative (as indicated by a negative probability) in response to an increase of the independent variable. Distributions for water addition compare watered and control plots; distributions for interspecific richness compare *Piper* species richness standardized as the proportion of the maximum richness used at a site; distributions for intraspecific richness compare low and high intraspecific richness treatments; distributions for insect richness compare responses per insect taxon present on an individual leaf.

The online version of this article includes the following figure supplement(s) for figure 2:

**Figure supplement 1.** Hierarchical Bayesian model (HBM) parameter estimates of percent herbivory, percentage of leaves with damage, variance in herbivory, and percent *Piper* survival against levels of water addition, *Piper* intraspecific richness, *Piper* interspecific richness, and insect richness at each site.

**Figure supplement 2.** Hierarchical Bayesian model (HBM) posterior parameter estimates of insect richness compared to levels of water addition, *Piper* intraspecific richness, and *Piper* interspecific richness.

## Analysis of plant survival

Water addition did not affect *Piper* survival in Costa Rica (z=−1.2, p>0.2), or in Ecuador (z=0.2, p>0.8), but survival was reduced by 48% in plots without water in Peru (z=3.2, p=0.001). Intraspecific richness increased survival by 38% in Costa Rica (z=−3.3, p<0.001) and by 32% in Ecuador (z=−4.9, p<0.001), but had no effect on survival in Peru (z=−1.55, p>0.1), Mogi (z=−0.22, p>0.8), or Uaimii (z=0.18, p>0.8). Interspecific richness had no effect on survival in Costa Rica (z=−1.5, p>0.1), Ecuador (z=−0.56, p>0.5), Peru (z=−0.49, p>0.6), or Uaimii (z=0.21, p>0.8). In Mogi, survival was reduced by 13% in plots with higher interspecific richness (z=−2.16, p=0.031). There was an interaction between intraspecific richness and water addition in Costa Rica and Peru. Survival in Costa Rica increased in response to water in high intraspecific richness plots, and decreased in response to water in low intraspecific richness plots (z=3.8, p<0.001), while the opposite pattern was observed in Peru (z=−2.4, p=0.02; *Figure 4—figure supplement 1*). Although plant die-offs cause a loss in richness in some plots, plant species identity was not related to survival with the exception of *Podophyllum peltatum*, which had the lowest survival rate of any species planted in Costa Rica. Further statistical results are available in *Supplementary file 1*.

## Discussion

Our results demonstrate two key patterns. First, the strength of effects of intraspecific richness on higher trophic levels is comparable to that of interspecific richness, supporting our predictions and corroborating recent studies demonstrating the importance of intraspecific richness (*Raffard et al., 2019*; *Koricheva and Hayes, 2018*; *Cook-Patton et al., 2011*). However, the direction of the effect of intraspecific richness on herbivory, insect richness, and plant survival can vary dramatically from the direction of effect of interspecific richness, in contrast to our predictions. Second, we found that perturbations in water availability can have complex effects on herbivores and plant survival, and that these effects can be modulated by plant diversity in a context-sensitive manner. While our prediction that water availability would influence herbivory across sites was supported, our results suggest that biodiversity loss and climatic perturbations may have dramatically different effects on ecosystem function at local scales, which may diminish our ability to predict how local communities will change as anthropogenic stressors increase.

As we did not directly measure plant stress or nutrition, it is difficult to determine the exact mechanism through which water addition reduced herbivory. The presence of an El Niño weather pattern during the course of the Peru experiment may have led the water addition treatment to relieve plants from drought stress, while water addition in Costa Rica and Ecuador may have added to water stress in treated plants as these sites received an above average level of rainfall during the course of the experiment. Despite this, water addition consistently suppressed herbivory in Costa Rica, Ecuador, and Peru under natural levels of *Piper* diversity, suggesting that predicted increases in precipitation in the next century (*Field et al., 2014a*) will dramatically alter the flow of resources from primary producers. Although we were only able to record the richness of insect herbivory patterns, this measure is indicative of the functional diversity of insect herbivores on *Piper* and changes to this value represent changes in interactions between *Piper* and higher trophic levels (*Dyer et al., 2010*; *Carvalho et al.,*

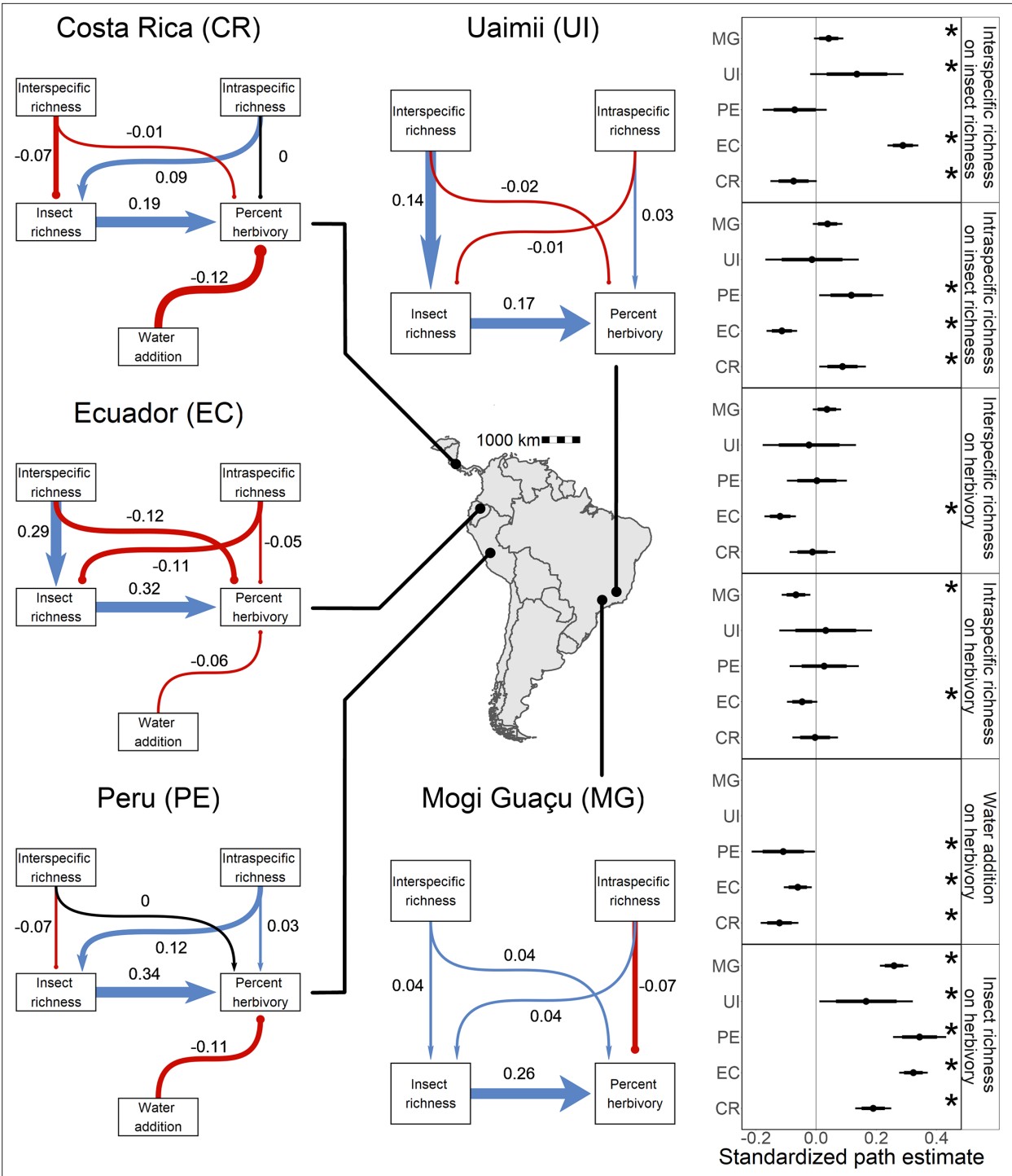

**Figure 3.** Direct and indirect effects of plant diversity and water availability on insect herbivores at five study sites. Bayesian structural equation models comparing effects of different drivers of herbivorous insect richness and herbivory at five sites. Standardized path coefficients are means of the posterior distribution for the effects estimated at each causal path. Positive relationships are indicated in blue with triangular heads, and negative relationships are indicated in red with circular heads. Black arrows indicate path coefficients of zero magnitude. Dot plots summarize the standardized mean of the posterior distribution for each causal path with 95% and 80% credible intervals. Asterisks indicate causal paths where the probability of an effect being positive or negative is >95%.

The online version of this article includes the following figure supplement(s) for figure 3:

**Figure supplement 1.** Bayesian structural equation models for drivers of insect richness, herbivory, and *Piper* survival at three sites, including interactions between intraspecific and interspecific richness, and water addition.

**Figure supplement 2.** Three causal models tested across sites.

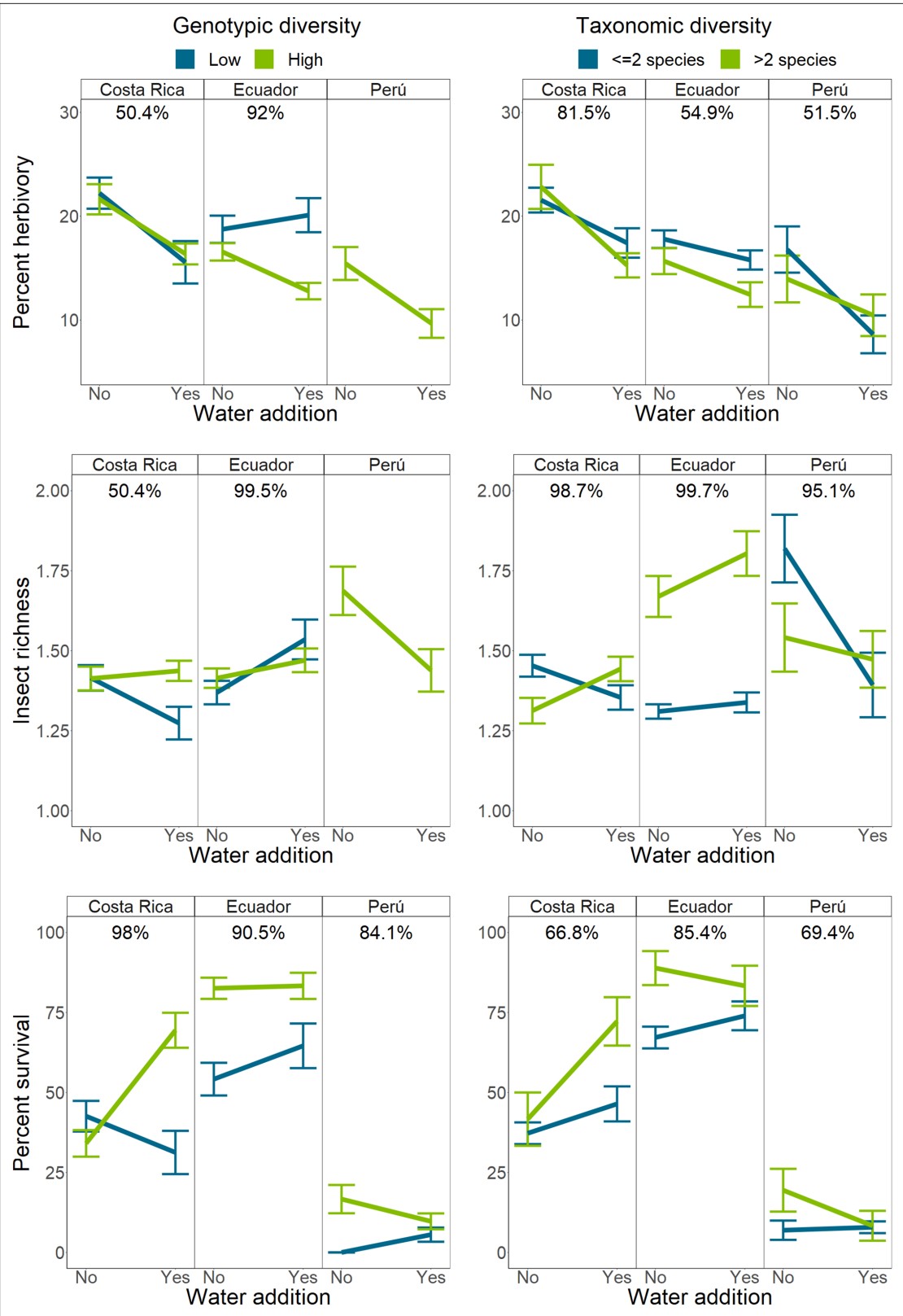

**Figure 4.** Interactions between intraspecific or interspecific richness and water availability on herbivory, herbivorous insect richness, and *Piper* survival. Bars indicate mean response and standard error of the mean. Percentages above each sub-figure indicate the probability that the two slopes are different as calculated using site-level hierarchical Bayesian models. Due to high mortality in Peru, interactions between water and intraspecific richness could not be compared for any responses except mortality. Single and two species richness plots have been combined for visualization purposes only.

*Figure 4 continued on next page*

*Figure 4 continued*

The online version of this article includes the following figure supplement(s) for figure 4:

**Figure supplement 1.** Percent *Piper* survival over time in five sites, compared to levels of intraspecific richness and water addition.

*2014*). As such, the additional reduction in effects of water addition on insect richness when *Piper* richness was low suggest that biodiversity loss in tropical systems will alter the ability of higher trophic levels to respond to environmental perturbations.

While our prediction that increased *Piper* interspecific richness would lead to increased insect diversity was met in the majority of sites, interspecific richness was associated with decreased insect diversity in Costa Rica and Peru. As herbivore taxa can be differently affected by manipulations of diversity (*Agrawal et al., 2006*), variation in the direction of the effect of intra- and interspecific richness may be due in part to variation in the composition of insect communities and herbivore pressure measured across sites. Changes in neighborhood effects when small numbers of plant species dominate a community can lead either to the reduction or increase of herbivore pressure, dependent both on the nature of plant species lost and on herbivore species present in a given community (*Barbosa et al., 2009*). As such, local variation in community composition has the potential to greatly alter the effects of both climate change and biodiversity loss on ecosystem function.

Although experimental methods varied somewhat between study sites, this cannot fully explain the level of heterogeneity observed in ecosystem response. For example, the methods employed in Costa Rica and Ecuador were nearly identical, and yet the directions of effect of intra- and interspecific diversity on insect richness were reversed in these sites (*Figure 2—figure supplement 2B and C*). There was considerable variation in both biotic (*Figure 1—figure supplement 2B*) and abiotic factors (*Supplementary file 2*) across sites, which may have contributed to the heterogeneity observed in ecosystem response to *Piper* diversity. Regardless of how biodiversity loss affects ecosystem function at large scales, variation in abiotic and biotic factors at locals scales can alter these effects, reducing our ability to predict how anthropogenic activity will alter ecosystem function. This is especially relevant in tropical systems, which have been the subject of far fewer studies of ecosystem function than temperate ecosystems. As such, our knowledge of local effects on the relationship between biodiversity and ecosystem function remains limited in these systems.

A long-standing question in ecology has been the extent to which ecosystem function increases with biodiversity and if this relationship plateaus at a level past which ecological redundancy predominates. Recent results from less complex temperate grassland systems suggest that these ecosystems can be described by a mostly linear relationship between richness and function, where even rare species make unique contributions to ecosystem function (*Isbell et al., 2011*). In these systems, high contingency can be expected, where ecosystem-level effects will depend on most of the interacting species. In contrast, we might expect that diverse, tropical communities could be characterized by greater ecological redundancy and thus be subject to less contingency (*Naeem, 1998*; *Rosenfeld, 2002*). Despite these expectations, our results demonstrate heterogeneity in ecosystem response to changes in both intraspecific and interspecific richness in five tropical sites, suggesting that complexity in these systems may not reduce the contingency effects of biodiversity loss. Understanding the impact of biodiversity loss in tropical forests is fundamental to our ability to conserve those systems, and our findings highlight the importance of approaching the study of changes in ecosystem function as context-sensitive responses in complex ecosystems.

## Materials and methods
### Study sites and focal plant genus

We conducted a large-scale transplant experiment replicated across five sites spanning 42° latitude in the Neotropics (*Figure 1*, *Supplementary file 2*) encompassing a range in annual precipitation from 1271 to 4495 mm (*Supplementary file 2*). At each site, we studied herbivory on planted individuals in the genus *Piper* (Piperaceae) in response to experimental treatments. Study sites included lowland equatorial humid forest at La Selva Biological Station, Costa Rica; high elevation equatorial humid forest in Yanayacu Biological Station, Ecuador; high elevation equatorial humid forest at El Fundo Génova, Peru; lowland seasonally dry gallery forest in the cerrado within the phytogeographic domain

of the Atlantic Forest in Mogi-Guaçu Biological Reserve, Brazil; and lowland seasonally semideciduous forest in the transition between the Atlantic Forest and the cerrado phytogeographic domains in Uaimii State Forest, Brazil. Climate classifications follow the Köppen-Geiger climate model (*Kottek et al., 2006*; *Supplementary file 2*).

Multiple species of *Piper* are found at all sites, ranging from 11 species in Mogi-Guaçu to 50 species in Costa Rica (*Salazar et al., 2016*). *Piper* is an ideal genus for large-scale comparative studies, as it is found across the Neotropics and subtropics, from ~10° N to about ~32° S. In addition to being widespread, *Piper* is abundant and diverse across its range, encompassing ~1000 species in the Neotropics (*Davidse et al., 2020*). *Piper* has been the subject of detailed studies of herbivory, and its herbivore fauna has been surveyed across its range (*Dyer and Letourneau, 1999*; *Dyer and Letourneau, 2003*; *Dyer et al., 2004*; *Letourneau et al., 2004*; *Connahs et al., 2009*; *Dyer et al., 2010*; *Bodner et al., 2012*; *Abarca et al., 2014*; *Glassmire et al., 2016*; *Slinn et al., 2018*; *Cosmo et al., 2019*; *Campos-Moreno et al., 2021*). Members of the genus host both specialist caterpillars and beetles as well as generalist caterpillars and Orthopterans (*Dyer and Palmer, 2004*; *Dyer and Letourneau, 1999*; *Dyer and Letourneau, 2003*; *Dyer et al., 2004*; *Letourneau et al., 2004*; *Dyer et al., 2010*). Leaf damage patterns produced by the different classes of herbivores are well documented and allow for the determination of unique taxa of herbivores (*Dyer et al., 2010*).

## Experimental design

Executing experiments across the Americas presents challenges, including nuanced variations in methodologies at each site. Nonetheless, the advantages of this expansive and consistent approach provides a greater understanding in the role of biodiversity in ecosystem function than examining isolated single-site studies. Here, we describe the experimental design applied across sites, see *Figure 1* and *Supplementary files 2 and 3* for site-specific details.

At each study site, a factorial experiment was implemented to test the effects of plant interspecific and intraspecific richness on herbivory, variation in herbivory, and insect richness. Experimental plants were propagated from cuttings of naturally occurring *Piper* plants (typically with three nodes and zero to one leaves). At each site a subset of naturally occurring *Piper* species was selected to act as a species pool for each experimental plot (*Supplementary file 3*). These species were selected to constitute a breadth of genetic and functional diversity representative of the *Piper* community present at each site. In all sites except Uaimii, experimental plots measured 4 m in diameter and contained 12 *Piper* individuals, planted either in pots with locally derived soil (for experiments manipulating water availability) or directly into the ground (for experiments without water additions in Mogi-Guaçu and Uaimii; *Supplementary file 2*) and cultivated without fertilizer or irrigation. Interspecific richness treatment levels consisted of single species monoculture plots, two species plots, and high richness plots with the maximum number of species available to produce cuttings (*Supplementary file 2*). The minimum number of *Piper* species in a high richness plot was 3 in Uaimii, and the maximum was 12 in Costa Rica and Ecuador. In plots with more than one species, species were randomly sampled from the species pool for that site.

An intraspecific richness treatment was crossed with the interspecific richness treatment. Intraspecific richness was manipulated by clones taken from a single mother plant (low richness) or individual cuttings all taken from unique mother plants (high richness). In the high intraspecific richness treatment, cuttings were from different *Piper* individuals growing at least 10 m apart, to eliminate the likelihood of a shared root system thus representing genetically unique individuals. In Costa Rica and Ecuador, the high interspecific richness plots only included individuals from unique mother plants because all cuttings were taken from different species.

A water addition treatment was crossed with the inter- and intraspecific richness treatments in Costa Rica, Ecuador, and Peru in order to examine how IPCC predicted regional mid-century increases in precipitation in these regionsmay affect herbivory and trophic interactions. Plants were cultivated in 5–6 L pots with drainage holes. Each potted plant under the water addition treatment was watered with 2 L of water twice per month in Costa Rica and Ecuador. In Peru, 2 L of water were added to plants every 3 months. Water was rapidly applied as a flooding event to completely saturate the soil. Plant cultivation periods lasted in excess of 1 year at all sites, and as such water was added in both the wet season and dry season. Experimental plots were randomly located in the study sites; in Mogi-Guaçu and Uaimii they were organized in three replicate blocks. There was a minimum of 20 m between

plots in blocks and 100 m between blocks. Randomly located plots were separated by a minimum of 50 m. Experimental periods lasted between 1.4 and 2.8 years, depending on the site (**Supplementary file 2**), and all measurements of herbivory were conducted on leaves that were initiated during these periods. Cuttings were replaced if they died during the first 3 months of the experiment. One of the authors was present at all of the sites for multiple visits to ensure as much standardization of treatment applications as possible.

*Piper* mortality resulted in a reduction in species richness in many plots and the loss of some treatment combinations, notably the loss of all *Piper* in unwatered, low intraspecific richness plots in Peru (**Figure 1—figure supplement 1B**). Plots originally planted as high interspecific richness treatments had a final richness of between 1 and 12 species in Costa Rica, 9 and 11 species in Ecuador, 1 and 3 species in Peru, 2 and 4 species in Mogi, and only 2 species in Uaimii.

To determine the effects of site variation in natural levels of precipitation on the outcomes of the water addition treatment, the absolute level of precipitation and precipitation anomalies relative to climate normals were collected for each month of the experimental periods in Costa Rica, Ecuador, and Peru (**Figure 1—figure supplement 3**). Climate data for Costa Rica were obtained from La Selva Biological Station. Data for Ecuador and Peru were obtained using TerraClimate (**Abatzoglou et al., 2018**), and an interpolation error in the precipitation for Peru in February 2016 was corrected using data from the National Service of Meteorology and Hydrology of Peru (SENAMHI).

## Measures of herbivory and insect richness

Plots were open to naturally occurring herbivores, and herbivory was recorded by taking photographs of all the experimental leaves at the end of the experimental period. Additional photos were taken every 3 months in Uaimii and Mogi-Guaçu, and in the first 5 months of the experiment in Costa Rica. These data were used to measure herbivory and to determine the types of herbivores feeding on leaves based on patterns of damage. When herbivores were observed on plants, they were photographed but were left to continue feeding so as to not interfere with the experiment. Herbivory was quantified by eye for each type of herbivore on each leaf by a single parataxonomist with extensive experience measuring herbivory on *Piper* following established protocols (**Dyer et al., 2010**). The amount of leaf area consumed was measured in relation to the total leaf area by visually dividing the leaf into equal sized segments to determine the percent area missing. This was measured as a continuous value to the greatest possible accuracy, typically 1–5% of the total leaf area. Insect herbivores were identified to the lowest taxonomic level possible based on their damage patterns (genus for specialist Lepidoptera, family for generalist Lepidoptera, family for Coleoptera, order for Orthoptera). Direct observations of herbivores were rare, so only damage patterns were used in analyses. As insect damage patterns on leaves are tightly correlated with insect richness in tropical forests (**Carvalho et al., 2014**), the different types of damage recorded were used as proxy for the richness of above ground insect herbivores on *Piper*. Hereafter, the term 'insect richness' refers to the richness of insect herbivore damage patterns on plants.

## Data analyses

The percentage of leaf tissue consumed by each insect taxon (based on damage patterns) on individual leaves was summed to determine the total percentage of herbivory on each leaf, and the variance in herbivory was calculated as the variance in leaf damage within each plant in a plot. Due to high mortality rates across sites, *Piper* interspecific richness in each plot was the final number of *Piper* species present in each plot at the end of the experiment (rather than the number of species planted) and was analyzed as a continuous covariate rather than a categorical treatment. Because most plant deaths occurred early in the experiments, final *Piper* interspecific richness more accurately reflects the local plant richness experienced by herbivores. Interspecific diversity for each plot was analyzed as the proportion of the interspecific diversity present in the species pools at each site. This enabled easier comparisons between intra- and interspecific richness, as intraspecific richness was only quantified as low and high, where high treatments represent the maximum intraspecific richness available at each site. The effect of intra- and interspecific diversity on each response variable is reported as the change in response as diversity changes from the lowest possible value at each site to the highest.

The effects of intraspecific richness, interspecific richness, and water addition on percent leaf area consumed, the percentage of damaged leaves, variance in herbivory, and insect richness were analyzed

across and within sites using HBMs. This framework acts as the Bayesian equivalent of a random-effects model where site is a random effect, allowing for generalized parameter estimates across sites. Analyses testing *Piper* intraspecific richness and interspecific richness were conducted using data from all sites; the effects of water addition and its interactions with richness were conducted using data from Ecuador, Costa Rica, and Peru. Due to high mortality in Peru, interactions between the water addition and intraspecific richness treatments could not be modeled for measures of herbivory or insect richness, and were only tested for survival.

We acknowledge that treatment-level combinations were not the same across the different sites, but experimental designs of this nature are encompassed within the framework of random-effects models, where different levels of random factors, such as site or year, consist of treatment levels that are unique to that site or year. This type of experimental design goes back to the origins of mixed models (**Fisher, 1919**; **Henderson et al., 1959**), and the lack of interactions between fixed and random effects increases generality in these models (**Abelson, 1995**). Even at a single site, manipulated variables in ecological experiments do not even consist of the same manipulations across all of the units of replication, as they suffer from problems such as multiple versions of treatments, interference, and noncompliance (**Kimmel et al., 2021**).

BSEMs were constructed for each site using all treatments as exogenous variables and insect richness and herbivory as endogenous variables. Three path models were developed under the assumptions that (1) intraspecific and interspecific richness may influence herbivory both directly and indirectly by modulating insect richness, (2) water addition may influence both herbivory and insect richness, and (3) insect richness may influence herbivory directly (**Table 1**). Models I-III were tested in Ecuador, Costa Rica, and Peru, while a model without the water addition variable was used to analyze data from Mogi-Guaçu and Uaimii (**Figure 3—figure supplement 2**). Additional models incorporating interactions between intraspecific richness, interspecific richness, and water availability were tested in Costa Rica, Ecuador, and Peru.

Models were run at the leaf level for herbivory and insect richness, and at the plant level for the percentage of leaves with damage, variance of herbivory, and *Piper* survival. For all HBMs and BSEMs, model convergence was estimated visually using traceplots and an $\hat{R}$ discriminatory threshold of 1.1 (**Gelman and Rubin, 1992**). Model fit was determined via PPCs using the sum of squares of the residuals as the discriminatory function (**Gelman et al., 1996**). A PPC near 0.5 indicates a high model fit, while values near one or zero indicate poor fit. BSEMs were further compared using the deviance information criterion.

All Bayesian models were written in JAGS via the jagsUI package in R (**Kellner et al., 2021**) using Markov chain Monte Carlo (MCMC) sampling with weakly informative priors. Residuals were modeled as normally distributed based on PPC comparisons between models. Models using gamma distributions for herbivory and binomial distributions for damage presence and mortality were found to consistently underestimate the magnitude of variance in the data based on PPC. For the majority of models, MCMC runs were conducted for 10,000 iterations using the first 1000 iterations as a burn-in phase to generate posterior distributions of parameter estimates for each response variable. HBMs modeling interactions required 20,000 iterations with the first 5000 as burn-ins for all models to consistently converge. Mean parameter estimates and 95% credible intervals (CIs) were calculated for all responses. 95% CIs which do not cross the y-axis are typically associated with less than a 2.5% type S error rate (**Gelman and Tuerlinckx, 2000**). A posterior PD was calculated based on the percentage of the posterior distribution responding in the same direction as the median response. A PD of 95%, for example, indicates that the same direction of response (e.g. a positive or negative response) was observed in 95% of iterations, regardless of the magnitude of the response (**Makowski et al., 2019**). PDs less than 95% indicate lower confidence that a relationship exists, but can still be interpreted as the probability that an effect exists.

To analyze the effects of the experimental treatments on plant survival, survivorship curves were constructed for all sites. Analyses of *Piper* survival were based on the initially planted interspecific richness treatment of each plot because mortality occurred early in the experiment. The effects of water addition, intra- and interspecific richness, and species identity on *Piper* survival were analyzed using Cox proportional hazard models. All data were analyzed using R software v4.1.0 (**R Development Core Team, 2013**).

## Acknowledgements

The authors would like to thank Hacienda San Isidro, Ecuador for access to their forest reserve for experiments, the Meteorological Service from Peru (SENAMHI) for access to their weather data, and IEF, Minas Gerais, for access to Uaimii Forest Reserve during experiments. The collecting permit in Peru was issued by Peruvian Ministry of Agriculture (RDG N 288-2015-SERFOR-DGGSPFFS). Funding was supplied through NSF grants DEB-1442103 to LAD, LAR, TLP, AMS, and DEB-1442075 to EJT.

## Additional information

### Funding

| Funder | Grant reference number | Author |
|---|---|---|
| National Science Foundation | DEB-1442103 | Lee A Dyer<br>Lora A Richards<br>Thomas Parchman<br>Angela M Smilanich |
| National Science Foundation | DEB-1442075 | Eric J Tepe |
| National Science Foundation | EN-2133818 | Lora A Richards |

The funders had no role in study design, data collection and interpretation, or the decision to submit the work for publication.

### Author contributions

Ari Grele, Data curation, Software, Formal analysis, Visualization, Writing - original draft, Writing – review and editing; Tara J Massad, Lee A Dyer, Eric J Tepe, Conceptualization, Funding acquisition, Investigation, Methodology, Writing – review and editing; Kathryn A Uckele, Formal analysis, Writing – review and editing; Yasmine Antonini, Laura Braga, Lidia Sulca, Humberto G Lopez, André R Nascimento, Investigation, Writing – review and editing; Matthew L Forister, Conceptualization, Writing – review and editing; Massuo Kato, Thomas Parchman, Angela M Smilanich, John O Stireman, Lora A Richards, Conceptualization, Funding acquisition, Methodology, Writing – review and editing; Wilmer R Simbaña, Data curation, Investigation, Writing – review and editing; Thomas Walla, Methodology, Writing – review and editing

### Author ORCIDs

Ari Grele (iD) http://orcid.org/0000-0002-0876-7682
Lidia Sulca (iD) http://orcid.org/0000-0002-1409-5827
Lora A Richards (iD) http://orcid.org/0000-0002-8052-4378

Reviewer #1 (Public Review): https://doi.org/10.7554/eLife.86988.3.sa1
Reviewer #2 (Public Review): https://doi.org/10.7554/eLife.86988.3.sa2
Author response https://doi.org/10.7554/eLife.86988.3.sa3

## Additional files

### Supplementary files

• Supplementary file 1. Mean parameter estimates and probability of direction (PD) for the effects of increases in intraspecific diversity, interspecific richness, water availability, and insect richness on measures of herbivory, plant mortality, and insect richness.

• Supplementary file 2. Study site characteristics and experimental details. *Climate data from La Selva Biological station are for the experimental period (2015–2018) and were provided by the Organization for Tropical Studies. Data from El Fundo Génova are based on nearby San Ramón (https://en.climate-data.org/south-america/peru/junin/san-ramon-28556/). Data from Ecuador were provided by Yanayacu Biological Station. Data for Mogi-Guaçu Biological Reserve are from January

2017 to December 2019 and are from the Centro Integrado de Informações Agrometeorológicas of São Paulo. Data for Uaimii State Forest are from the Plano de Manejo FLOE Uaimii, Instituto Estadual de Florestas of Minas Gerais

- Supplementary file 3. Species of *Piper* used at each study location. * indicates morphospecies.
- MDAR checklist

## Data availability

Data and code used for all analyses and figures can be accessed via the Dryad repository at https://doi.org/10.5061/dryad.8w9ghx3rf.

The following dataset was generated:

| Author(s) | Year | Dataset title | Dataset URL | Database and Identifier |
|---|---|---|---|---|
| Ari G, Tara M, Kathryn U, Lee D, Yasmine A, Laura B, Matthew F, Lidia SG, Massuo K, Humberto G, André N, Tom P, Wilmer S, Angela S, John S, Eric T, Thomas W, Lora R | 2024 | Intra and interspecific diversity in a tropical plant clade alter herbivory and ecosystem resilience | https://doi.org/10.5061/dryad.8w9ghx3rf | Dryad Digital Repository, 10.5061/dryad.8w9ghx3rf |

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
