## [Editor Report · eLife assessment]

This **important**, large experimental study examines the effects of plant species richness, plant genotypic richness, and soil water availability on herbivory patterns for Piper species in several tropical sites. The authors find **solid** evidence that water availability, as well as intra- and interspecific plant diversity, influence herbivory and herbivore diversity, but that the effects differ geographically.

---

## [Referee Report · Reviewer #1 (Public Review)]

This study reports a long-term, multisite study of tropical herbivory on Piper plants. The results are clear that lack of water leads to lower plant survival and altered herbivory. The results varied substantially among sites. The caveats are that ecosystem processes beyond water availability are not investigated although they are brought into play in the title and in the paper, that herbivory beyond leaf damage was not reported (there might be none, reader needs to be shown the evidence for this), that herbivore diversity is defined by leaf damage (authors need to give evidence that this is a valid inference), that the plots were isolated from herbivores beyond their borders, that the effects of extreme climate events were isolated to Peru, that intraspecific variation in the host plants needs to be explained and interpreted in more detail, the results as reported are extremely complicated, the discussion is overly long and diffuse.

---

## [Referee Report · Reviewer #2 (Public Review)]

This is an important and large experimental study examining the effects of plant species richness, plant genotypic richness, and soil water availability on herbivory patterns on Piper species in tropical forests.

A major strength is the size of the study and the fact that it tackled so many potentially important factors simultaneously. The authors examined both interspecific plant diversity and intraspecific plant diversity. They crossed that with a water availability treatment. And they repeated the experiment across five geographically separated sites.

The authors find that both water availability and plant diversity, intraspecific and interspecific, influence herbivore diversity and herbivory, but that the effects differ in important ways across sites. I found the study to be solid and the results to be very convincing. The results will help the field grapple with the importance of environmental change and biodiversity loss and how they structure communities and alter species interactions.

---

## [Author Response]

The following is the authors’ response to the original reviews.

**Reviewer #1 (Public Review):**

We thank reviewer #1 for identifying the major caveats of the paper, and have split them out into separate comments below to address them.

Comment (1) The caveats are that ecosystem processes beyond water availability are not investigated although they are brought into play in the title and in the paper

Author response: We disagree that water availability is the only ecosystem process investigated in this study, as herbivory, plant mortality, and the maintenance of diversity in higher trophic levels are important processes within ecosystems. We have added text to the abstract and introduction clarifying that we consider these response measures to be ecosystem processes. Further language to this effect already exists in the abstract, methods, and discussion.

Comment (2) That herbivory beyond leaf damage was not reported (there might be none, the reader needs to be shown the evidence for this)

Author response: This is typically how herbivory is assessed in ecological studies, and our focus is on folivores. There may be additional herbivory in the form of fluid-sucking insects, shoot/root herbivory, etc., but these were not assessed. It would be interesting to assess these other forms of herbivory to see if they respond similarly with additional studies.

Comment (3) That herbivore diversity is defined by leaf damage (authors need to give evidence that this is a valid inference)

Author response: We thank reviewer #1 for pointing out the lack of written support for this claim. We have modified the methods (lines 138-139; 214-217) to clarify that this is a useful proxy for insect richness in the Piper system, and have added citations demonstrating it has been found to correlate well with insect richness in tropical forests.

Comment (4) That the plots were isolated from herbivores beyond their borders

Author response: This was not an assumption of the study. We have modified the methods (line 200) to make this clearer to the reader.

Comment (5) That the effects of extreme climate events were isolated to Peru

Author response: This was not an assumption of the study, rather it is an observation. While we consider it important to include observed climate differences between sites in the interpretation of our results, it was not necessary for there to be extreme climate events at other sites as we consider manipulated water availability to represent changes in precipitation that are expected to occur at these sites with climate change.

Comment (6) That intraspecific variation in the host plants needs to be explained and interpreted in more detail

Author response: We thank reviewer #1 for identifying that our current explanations needed development. We have modified the introduction to explore potential mechanisms relating intraspecific diversity to ecosystem function based on recent studies, and have modified the discussion to bring focus to why the effects of intraspecific differ from interspecific.

**Reviewer #1 (Recommendations For The Authors):**
Comment (1) Pare this material down to simpler results. The most significant to me is the intraspecific variation in damage. Were this broken out and reported in some detail it could be quite interesting. I find the results to be a confusing blizzard of multiple factors that differ among sites; after reading the paper twice I could not recall the takeaway lesson beyond that drought wrecks the diversity of herbivores and sometimes even kills the host plant.

Author response: We agree that the results are complicated given the variation in effects among sites, but this variation and complexity is important – and is in itself is one of the takeaway points. Unfortunately, nature is not simple. We have made several large edits to the results section, including the removal of methodological and otherwise redundant information, to hopefully bring the major takeaways into focus.

**Reviewer #2 (Public Review):**
Comment (1) This is an important and large experimental study examining the effects of plant species richness, plant genotypic richness, and soil water availability on herbivory patterns on Piper species in tropical forests.A major strength is the size of the study and the fact that it tackled so many potentially important factors simultaneously. The authors examined both interspecific plant diversity and intraspecific plant diversity. They crossed that with a water availability treatment. And they repeated the experiment across five geographically separated sites.The authors find that both water availability and plant diversity, intraspecific and interspecific, influence herbivore diversity and herbivory, but that the effects differ in important ways across sites. I found the study to be solid and the results to be very convincing. The results will help the field grapple with the importance of environmental change and biodiversity loss and how they structure communities and alter species interactions.

Author response: We thank reviewer #2 for their kind words.

**Reviewer #2 (Recommendations For The Authors):**
Comment (1) I was confused about why the authors measured species diversity/richness as a proportion of the species pool. This means that the metric of richness decreases if species are added to the species pool but not the plot/experiment. I think I understand it, but I suggest the authors explain this choice.

Author response: We thank reviewer #2 for pointing out that this was confusing. We have clarified the methods (lines 228-232) to explain that this choice was made to allow easier comparison between intra- and interspecific richness.

Comment (2) One of the stronger estimated relationships was a positive effect of plant species richness on insect richness. I found it a little hard to interpret this relationship. Is this just because there are host species specialists? So, with more host species there are more herbivore species? Or does insect richness increase multiplicatively with increasing plant species richness? One way to look for this would be for the authors to examine the relationship between plant species richness and the average number of herbivore damage types per plant species.

Author response: We agree that this is important for the reader to understand and have added text to the introduction and discussion sections explaining that this is the expectation based on theory and other empirical studies. We have additionally added text to the discussion (lines 386-388) pointing out that this pattern was not observed at all sites. While we agree that it would be interesting to explore if this effect was additive or multiplicative, we do not believe this is in the scope of the paper due to the methods used to measure insect richness.

Comment (3) Unless I missed it, some important information about the models was missing. E.g., what distributions were assumed for each of the variables? Any transformations?

Author response: We thank reviewer #2 for pointing this out, this information has been added to the methods (lines 272-274)

Comment (4) Why is there no model with water addition affecting insect richness directly but not percent herbivory directly?

Author response: While we originally decided to not include this model due to lack of theoretical support and low statistical performance, we have added references to this model (now model II) in the methods and results for consistency and to make model performance clearer to the reader. We have additionally moved supplemental table S1 to the main text to make the models and hypotheses tested by each model more accessible.

Comment (5) Fig. 2. What are the percentages above the figures? Maybe PD values?

Author response: These values are now clarified in the figure caption

Comment (6) L364 "can differ dramatically" This is vague and confusing. Differ in what way? From each other? Did the authors really expect plant richness to have the same effect on herbivory and plant survival? What would it mean anyway for plant richness to have the same effect on herbivory and plant survival?

Author response: We agree that the language here is confusing and thank reviewer #1 for drawing our attention to it. We have modified the discussion (lines 363-365) to clarify that the direction of effect of intraspecific richness can vary from the direction of effect of interspecific richness, rather than the effects on different response variables varying from each other.

Comment (7) L 375 "only meaningful differences" This statement feels a little overly strong. It seems like there is a good argument for this, but there could be other things going on.

Author response: We agree that the language here was unnecessarily strong, and have modified the discussion (lines 398-403) to focus on the lack of difference between methodologies at these two sites, and the observed differences in climate and community structure at each site.